# Small RNA and Degradome Sequencing in Floral Bud Reveal Roles of miRNAs in Dormancy Release of *Chimonanthus praecox*

**DOI:** 10.3390/ijms24044210

**Published:** 2023-02-20

**Authors:** Ning Liu, Yingjie Jiang, Ting Zhu, Zhineng Li, Shunzhao Sui

**Affiliations:** Chongqing Engineering Research Center for Floriculture, Key Laboratory of Horticulture Science for Southern Mountainous Regions of Ministry of Education, College of Horticulture and Landscape Architecture, Southwest University, Chongqing 400715, China

**Keywords:** *Chimonanthus praecox*, floral bud dormancy, high-throughput sequencing technology, miRNAs, target identification

## Abstract

*Chimonanthus praecox* (wintersweet) is highly valued ornamentally and economically. Floral bud dormancy is an important biological characteristic in the life cycle of wintersweet, and a certain period of chilling accumulation is necessary for breaking floral bud dormancy. Understanding the mechanism of floral bud dormancy release is essential for developing measures against the effects of global warming. miRNAs play important roles in low-temperature regulation of flower bud dormancy through mechanisms that are unclear. In this study, small RNA and degradome sequencing were performed for wintersweet floral buds in dormancy and break stages for the first time. Small RNA sequencing identified 862 known and 402 novel miRNAs; 23 differentially expressed miRNAs (10 known and 13 novel) were screened via comparative analysis of breaking and other dormant floral bud samples. Degradome sequencing identified 1707 target genes of 21 differentially expressed miRNAs. The annotations of the predicted target genes showed that these miRNAs were mainly involved in the regulation of phytohormone metabolism and signal transduction, epigenetic modification, transcription factors, amino acid metabolism, and stress response, etc., during the dormancy release of wintersweet floral buds. These data provide an important foundation for further research on the mechanism of floral bud dormancy in wintersweet.

## 1. Introduction

MicroRNAs (miRNAs), the key post-transcriptional regulators of eukaryotic gene expression, are a class of endogenous non-coding small RNAs (sRNAs) with 18–24 nucleotides (nt) in length [1]. Mature miRNAs are processed from larger primary miRNAs (pri-miRNAs) with a series of endonucleases and then associated with Argonaute (AGO) proteins to form RNA-induced silencing complexes (RISCs). Based on perfect or near-perfect complementarity between miRNAs and target mRNAs, RISCs negatively regulate gene expression by targeting mRNAs for cleavage or translational repression [2,3]. To date, a large number of miRNAs have been identified in various plant species, such as rice, maize, Arabidopsis, poplar, grape, and peach [4,5,6,7,8,9]. Furthermore, numerous studies have shown that miRNAs play important regulatory roles in various biological processes, including plant growth and development, signal transduction, and biotic and abiotic stress [10,11,12,13]. miRNAs show both conservation and diversity among plant lineages. Some miRNAs are conserved in angiosperms and even embryophyta, whereas more miRNAs show species specificity, which reflects the natures of rapid evolution and functional differentiation of miRNAs [14,15].

Bud dormancy is a temporary cessation of visible growth during plant growth and development, and it is a biological characteristic of perennial deciduous plants adapted to environmental and seasonal changes during long-term evolution [16,17,18]. According to the factors that cause dormancy, bud dormancy is generally classified into three types: paradormancy, endodormancy, and ecodormancy [16]. Endodormancy, also known as natural dormancy, is controlled by endogenous factors of dormancy structure. Short days (SDs) and/or low temperatures are the main factors for inducing bud endodormancy, and endodormant buds do not resume growth until the chilling requirement is fulfilled [19]. In this study, endodormancy is referred to as dormancy unless specified otherwise.

Bud dormancy is a complex physiological process regulated by many internal and external factors. Bud dormancy plays crucial roles in the growth and development, flowering and fruiting, and species survival of perennial deciduous plants [20]. In the past two decades, the central role of *short vegetative phase* (*SVP*), mostly called *dormancy-associated MADS-box* (*DAM*) in Rosaceae, in regulating bud dormancy has been extensively documented [21,22,23,24,25,26,27,28]. Plant hormones, such as abscisic acid (ABA), gibberellin (GA), auxin (indole-3-acetic acid, IAA), ethylene (ET), jasmonates (JAs), etc., are important internal mediators in the regulation of the bud dormancy cycle. *SVP/DAM* can integrate the metabolism and signal transduction of ABA and GA to regulate bud dormancy [18,26,29]. In recent years, the roles of epigenetic modifications, including DNA methylation, histone modification, and chromatin remodeling, in regulating bud dormancy have been reported [18,30,31,32,33,34,35,36]. And epigenetic modification is considered to be a main mechanism that regulates the *SVP/DAM* gene during dormancy [18,29]. In peach cultivars, changes in histone H3 lysine-4 trimethylation (H3K4me3), histone H3 lysine-27 trimethylation (H3K27me3), and histone H3 acetylation (H3ac) of *DAM6* were closely related to specific chilling requirements and the date of endodormancy release [30]. The reduction in H3K4me3 and H3ac modification of *AcSVP2* resulted in the downregulation of *AcSVP2* and endodormancy release during kiwifruit winter dormancy [31]. miRNAs are an important epigenetic modification; however, their regulations in bud dormancy are scarcely reported. Only several studies have identified miRNAs that regulate bud dormancy in trees, including pear, apple, peony, peach, and grape [21,37,38,39,40]. In white pear, miRNA6390 directly targets *PpDAM* to inhibit its expression, thereby promoting the release of floral bud dormancy [21]. Furthermore, miR156k, miR159a, miR167a, etc., were found to be differentially expressed in tree peony buds between endodormancy and ecodormancy [38]. In peach, miR6285 was found to regulate dormancy release in flower buds by targeting *asparagine-rich protein* (*NRP*), which is involved in ABA signal transduction [39]. The miR159-MYB module could be involved in regulating apple bud dormancy by mediating the homeostasis of endogenous ABA [37]. Most of these data were obtained with high-throughput sequencing; thus, the regulation mechanisms of miRNAs in bud dormancy are unclear.

*Chimonanthus praecox* (wintersweet), a perennial woody ornamental plant, is a rare precious flower and cut flower material in winter that has high ornamental and economic value. In autumn, wintersweet floral buds are induced to enter dormancy, and a certain period of chilling accumulation is essential to break dormancy [41,42]. In recent years, the deficiency of chilling accumulation caused by global warming has interrupted normal dormancy release in some cases, which seriously affects the flowering process and quality of wintersweet. Although some studies on the molecular mechanisms of wintersweet floral bud dormancy regulation were performed at the transcriptomic and proteomic levels [41], the roles of miRNAs involved in chilling-mediated dormancy control of wintersweet floral buds remain unclear. High-throughput sequencing technology and bioinformatics are effective in identifying miRNAs. Therefore, in this study, sRNA sequencing (sRNA-seq) of wintersweet floral buds was performed to identify miRNAs involved in the regulation of dormancy release through next-generation sequencing. Meanwhile, the target genes of these miRNAs were identified with degradome sequencing (parallel analysis of RNA end sequencing, PARE-seq). This study provides a basis for further analyzing the roles of miRNAs during the dormancy of wintersweet floral buds and also provides a platform for investigation of the biological functions of miRNAs in wintersweet. In addition, our work contributes to a better understanding of the mechanism of bud dormancy in perennial plants.

## 2. Results

### 2.1. Overview of sRNA Sequencing

With *C. praecox* ‘Concolor’ plants as the materials, six samples of floral buds (FBs) (FB.Nov, nF16, FB150, FB300, FB450, and IB570, three biological replicates for each sample) with different chilling accumulation during the dormancy process were used for sRNA sequencing, and 18 sRNA libraries were constructed. These libraries generated a total of 358,543,196 clean reads through high-throughput sequencing and preliminary filtering. After removing contamination, low-quality reads, and reads with length <18 nt, 324,509,314 clean tags were obtained, including 39,956,685 unique tags (non-redundant sequences). The sequence length of these sRNAs was concentrated at 20–24 nt, accounting for 86.92% of clean tags. For all samples, 21- and 24-nt sRNAs were the most abundant classes (Figure 1). These lengths are consistent with the specificity of Dicer-like (DCL1–4) proteins that cleave double-stranded RNA (dsRNA) to form sRNA [43], which is consistent with the length distribution of plant sRNAs. Among all samples, 24-nt sRNAs were the most abundant in FB.Nov, FB150, and FB450, whereas 21-nt sRNAs were the most abundant in the other three samples (Figure 1). The results showed that the sRNAs’ sequences were diverse and complex in these samples.

These clean tags were mapped to the GenBank, Rfam, RepeatMasker, and *C. praecox* transcriptome databases successively for annotation and classification. A total of 11.45% and 22.87% of clean tags were annotated as non-coding structural RNAs (rRNA, tRNA, snRNA, and snoRNA) and transcriptome sequences (degraded fragments of wintersweet mRNAs), respectively (Table 1). The annotated tags were removed, and the remaining clean tags were further analyzed to identify known and novel miRNAs of wintersweet.

### 2.2. Identification of Known and Novel miRNAs

All unannotated clean tags were aligned with known plant miRNAs sequences in miRBase 21.0. A total of 862 known miRNAs were identified in all six samples (18 libraries) of wintersweet (Appendix A), containing 74,164,064 tags and accounting for 22.85% of total clean tags (Table 1). These known miRNAs belonged to 754 families (Appendix A). The expression profiles of these known miRNAs showed great differences in the amounts of tags. Among them, the total abundance of miR166-y was the highest, with 31,649,892 tags from all libraries, followed by miR396-x (27,464,586). However, 79.70% of known miRNAs had fewer than 10 tags (Appendix A). The results indicated that the expression of these known miRNAs could have spatiotemporal specificity, and that these known miRNAs might play different regulatory roles in the dormancy of wintersweet floral buds. In addition, only 27 precursors and their secondary structures of 19 miRNAs from 862 known miRNAs were identified in the *C. praecox* transcriptome database, which included the same precursors forming mature miRNAs with different sequences and mature miRNAs with the same sequences derived from different precursors (Appendix A). The main reasons for this result were the lack of both an available genome database and a complete transcriptome database for wintersweet.

Further, the tags annotated as known miRNAs were excluded, and the remaining clean tags were aligned against the *C. praecox* transcriptome database to identify the loci that may serve as precursors (pre-miRNA) of miRNAs. By predicting and analyzing the hairpin structures of these putative precursors, 402 potential novel miRNAs were identified in all samples of wintersweet (Appendix A), containing 1,014,923 tags and accounting for 0.31% of the total clean tags (Table 1). Compared with known miRNAs, the expression levels of novel miRNAs were relatively low (Appendix A). The precursors of all 402 novel miRNAs possessed typical stem-loop structures. These precursors were 57–371 nt in length; the minimum folding free energy (MFE) of these precursors hairpin structures ranged from −166.5 kcal·mol^−1^ to −18.0 kcal·mol^−1^ (Appendix A), which was similar to the characteristic of other plant miRNAs precursors [44,45]. Among the precursors of 402 novel miRNAs, 39 precursors formed mature miRNAs on both 3′ and 5′ arms (Appendix A). In addition, similar to known miRNAs, some mature novel miRNAs with the same sequence were derived from different precursors, such as novel-m0010 and novel-m0011 (Appendix A). Secondary structure prediction of several precursors is shown in Appendix A.

Sequence analysis of all of the 1264 miRNAs (862 known and 402 novel) revealed that their lengths ranged from 18 nt to 26 nt. miRNAs with 21 nt were the most abundant, accounting for 85.50% of the total miRNAs, followed by 22- (7.12%) and 20-nt (7.00%) miRNAs, whereas only one 26-nt miRNA was identified (Figure 2a). Studies have shown that the base composition of miRNAs can affect their physicochemical and biological properties [46]. When AGO proteins bind to miRNAs to form RISCs, different AGO proteins are preferentially enriched for the different initiating nucleotides of miRNAs [47]. The analysis of first nucleotide bias showed that the first nucleotide of 37.82% (478) and 30.06% (380) of miRNAs were U and A, respectively. For miRNAs of different lengths, the first nucleotides of 20-, 21-, and 22-nt miRNAs were biased toward U, whereas those of other miRNAs were biased toward A (Figure 2b).

### 2.3. Differential Expression of miRNAs during Dormancy Release

The expression levels of all identified miRNAs were calculated by using the transcripts per million (TPM) algorithm. According to the expression levels of the miRNAs in each sample, all samples were compared in pairs to screen the differentially expressed miRNAs (DEMs) between different samples. A total of 301 DEMs were obtained (Appendix A, Appendix A). The results from the statistical analysis of DEMs between different samples are shown in Figure 3a, which reveals that the differences in miRNAs expression became more significant with chilling accumulation; notably, the differences between the dormancy breaking sample (IB570) and samples remaining in dormancy (FB.Nov, nF16, and FB150/300/450) were the greatest.

To identify miRNAs involved in the regulation of wintersweet floral bud dormancy release, we focused on the comparative analysis of DEMs among IB570/FB.Nov, IB570/nF16, IB570/FB150, IB570/FB300, and IB570/FB450, and 23 DEMs shared in the five comparisons were screened, including 10 known and 13 novel miRNAs (Figure 3b, Appendix A). Of the 10 known miRNAs, miR157-x, miR167-x, miR167-y, miR390-y, and miR395-y were highly conserved and were studied in various plants; miR473 was reported only in several tree species (*Populus trichocarpa*, *Prunus persica*, and *Citrus trifolia*) [9,49,50]; the other miRNAs (miR2092, miR5179, miR6293, and miR9566) were rarely reported. A heatmap depicting the expression of the 23 DEMs showed that 15 miRNAs (5 known and 10 novel) were upregulated, and 8 miRNAs (5 known and 3 novel) were downregulated in IB570 compared with other samples (FB.Nov, nF16, and FB150/300/450) (Figure 3c). Among these DEMs, only miR167-x had very high expression (TPM > 20,000) in all samples, whereas the expression levels (TPM) of the remaining miRNAs were <100 (Appendix A), indicating that miR167-x plays an important regulatory role in the induction and release of dormancy in wintersweet floral buds. miR6293-x and miR9566-y were barely expressed in all samples with remaining dormancy, except IB570 (Appendix A), which indicated that miR6293-x and miR9566-y might promote dormancy release in wintersweet floral buds. With the continuous accumulation of chilling, the expression of novel-m0186-3p showed an increasing trend, whereas the expression of novel-m0077-5p showed a decreasing trend compared with that of FB.Nov (Appendix A), indicating that novel-m0186-3p and novel-m0077-5p could respond to a chilling environment and have opposite roles in regulating the dormancy of wintersweet floral buds.

**Figure 3 ijms-24-04210-f003:**
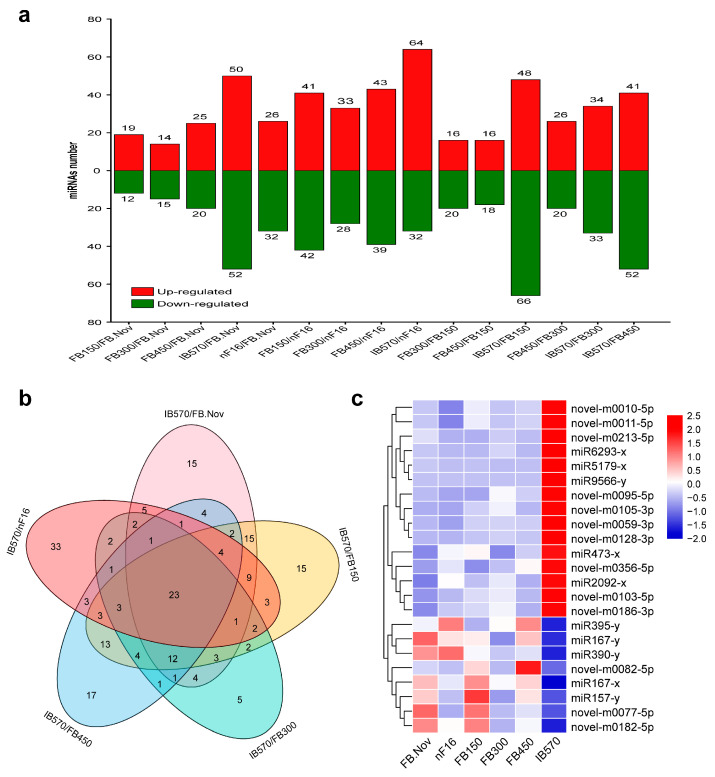
Differentially expressed miRNAs (DEMs). (**a**) The distribution of DEMs. (**b**) Venn diagram of DEMs from IB570/FB.Nov, IB570/FB150, IB570/FB300, IB570/FB450, and IB570/nF16. The numbers in the figure represent the number of DEMs. (**c**) Heatmap of the 23 DEMs. Red and blue represent upregulation and downregulation, respectively. (**a**,**b**) were completed with TBtools [51] software with default parameters. (**c**) was completed with TBtools software; the values of miRNA expression were normalized using Z-score normalization.

### 2.4. Summary of Degradome Sequencing

Equal amounts of total RNA from each sample were mixed for degradome sequencing, and one degradome library was constructed. After sequencing and data filtering were completed, 37,913,833 clean tags were obtained, including 12,290,544 unique tags. After removing non-coding structural RNAs and polyN fragments, the remaining clean tags were aligned against the *C. praecox* transcriptome database. A total of 31.33% of the unique tags were positively aligned with the database (annotated as cDNA_sense), and 29.14% of the unique tags were reverse complemented to the database (annotated as cDNA_antisense) (Table 2). Tags annotated as cDNA (cDNA_sense and cDNA_antisense) were used for the detection of miRNAs target genes.

### 2.5. Identification of miRNA Targets

Taking the *C. praecox* transcriptome database as reference, the cleavage sites of 1264 miRNAs identified with the described sRNA-seq were detected. A total of 220,567 cleavage sites of 1186 miRNAs were detected, corresponding to 33,310 degradation genes and 230,989 miRNA–target pairs (Appendix A). According to the abundance of tags in degradome sequencing, these miRNA–target pairs were divided into five categories: category 0, 1, 2, 3, and 4, which included 1374, 1312, 98,246, 8170, and 121,887 miRNA–target pairs, respectively (Figure 4a, Appendix A). Based on the detection results of cleavage sites, target plots (T-plots) were drawn in units of genes. The T-plot of two target genes is shown in Figure 4c,d.

Degradome sequencing identified 1707 genes as target genes of 21 miRNAs (9 known and 12 novel) from 23 DEMs, with the exception of miR473-x and novel-m0105-3p (Appendix A), including 1940 miRNA–target pairs (Appendix A). These miRNA–target pairs were mainly in category 2 (46.14%) and category 4 (49.07%), whereas the most reliable categories, category 0 and 1, only contained 6 (0.34%) and 5 (0.28%), respectively (Figure 4b), which necessitates further experimental validation. Among the 21 DEMs, novel-m0077-5p, novel-m0182-5p, and novel-m0103-5p targeted 2, 9 genes, and 1 gene, respectively, whereas the number of target genes of the remaining miRNAs ranged from 20 to 458. In addition, miRNAs with different sequences could target the same gene (e.g., miR167-x and miR2092-x targeted *ELF6*; novel-m0059-3p and novel-m0213-5p targeted *TIR1* and *SPL15*) (Appendix A). These results suggest that these miRNAs have multiple regulatory roles, and different miRNAs may synergistically regulate the same target gene.

A total of 1532 of the 1707 target genes identified were annotated with corresponding functions. Among them, most encode metabolism-related enzymes or proteins that are involved in amino acid metabolism, epigenetic regulation (DNA methylation, histone modification, chromatin remodeling, and RNA modification, etc.), phytohormone metabolism and signal transduction (ABA, IAA, and ET, etc.), glucose metabolism, and stress response. Some target genes are related to flower development (such as *FUL-like*, *SEP1*, *COL5*, and *FCA*) and SPL, MYB, WRKY, and bHLH transcription factors. In addition, there are also a large number of target genes encoding structural component domains of certain proteins (Appendix A). These results indicate that miRNAs are involved in the regulation of various biological processes during the dormancy release of wintersweet floral buds.

Among the 1707 target genes, Gene Ontology (GO) enrichment analysis was performed on targets from miRNA-target pairs that were classified into categories 0, 1, and 2 with high reliability; unfortunately, only 320 of 855 genes were annotated. In Cellular component, significantly enriched GO terms were only found in the endomembrane system, microtubule associated complex, phosphatase complex, and protein serine/threonine phosphatase complex. The Molecular functions were mainly related to transition metalion binding, transferase activity, transferring glycosyl groups, isomerase activity, and anion binding. Regarding Biological process, these target genes were mainly involved in single-organism process, single-organism cellular process, single-organism metabolic process, biological regulation, small molecule metabolic process, cellular component organization or biogenesis, and cellular component organization (Figure 5, Appendix A). The GO enrichment analysis provided abundant resources for target gene isolation, localization, and expression profiling in wintersweet.

## 3. Discussion

Wintersweet is a valuable flowering plant in winter. A certain period of chilling accumulation is necessary to break its floral buds’ dormancy and promote flowering. miRNAs are important post-transcriptional regulators involved in various biological processes in plants. Here, sRNA sequencing and degradome sequencing were combined to study the roles of miRNAs in the chilling-mediated dormancy release of wintersweet floral buds for the first time.

In the present study, the transcriptome data of *Chimonanthus praecox* was used for the annotation of miRNAs and their target genes rather than the genome, though two genomes were published [52,53]. The main reasons for this include the following two aspects: the annotation information of both genomes that were deposited in the NCBI (National Center for Biotechnology Information) is lacking; moreover, there may be some conflicts of interest due to the copyright issues involved.

In this study, sRNA sequencing revealed that in all six samples, 21- and 24-nt sRNAs were the two most abundant sRNAs. Of these, 24-nt sRNAs were the most abundant in FB.Nov, FB150, and FB450, whereas 21-nt sRNAs were the most abundant in the other three samples. These results were similar to those obtained in *Lilium pumilum* [54]. However, many plant studies have shown that 24-nt sRNAs are dominant in sRNA libraries. For instance, studies on winter wheat (*Triticum aestivum* L.), tea plant (*Camellia sinensis*), tree peony (*Paeonia suffruticosa* Andrews), trifoliate orange (*Citrus trifoliata*), and Chonglou (*Paris polyphylla* var. *yunnanensis*) have shown that 24-nt sRNAs are the most abundant, followed by 21-nt sRNAs [10,38,55,56,57]; in maize (*Zea mays*), the most abundant sRNA is 24 nt, followed by 22 nt [58]; in Japanese pear (*Pyrus pyrifolia* “Kosui”), hot pepper (*Capsicum annuum* L.), and turnip rape (*Brassica rapa* L.), 24-nt sRNAs are the most abundant, followed by 23-nt sRNAs [45,46,59]. By contrast, 21-nt sRNAs are most abundant in apple (*Malus domestica* cv.) and masson pine (*Pinus massoniana*) [11,37]. These results suggest that sRNA transcripts are complex and vary significantly in different plants.

In plants, 21-nt miRNAs are typical and play important roles in post-transcriptional gene silencing by mediating mRNA cleavage or translational repression [43]. Of the 1264 miRNAs (862 known and 402 novel) identified in this study, 21-nt miRNAs accounted for 85.50%, which was consistent with the report that most plant miRNAs are 21 nt long [60]. The analysis of first nucleotide bias showed that the first bases of 20-, 21-, and 22-nt miRNAs were biased toward U, and those of other miRNAs with different lengths were biased toward A (Figure 2). AGO proteins are the core components of RISCs and have certain bias toward the starting nucleotide of miRNAs when they associate with miRNAs to form RISCs. In *Arabidopsis*, AGO1 predominantly binds miRNAs that initiate with base U; AGO2 and AGO4 preferentially recruit miRNAs with 5′ terminal A; whereas AGO5 is biased toward miRNAs with 5′ terminal C [46,47,54]. These findings indicate that the nucleotide at the 5′ terminal of a miRNA is closely related to its biological activity, and if it is changed, the miRNA may interact with one different AGO protein and show different biological functions.

In plants, nonconserved miRNAs are generally expressed at low levels or mainly expressed in specific cells or under specific growth conditions [7]. In this study, the abundance of novel miRNAs was generally lower than that of known miRNAs, which is consistent with the results of various plants reported so far, such as those from analyses of the Japanese pear (*Pyrus pyrifolia* “Kosui”), turnip rape (*Brassica rapa* L.), and hot pepper (*Capsicum annuum* L.) [45,46,59]. Furthermore, according to the miRBase database, these novel miRNAs have not been reported in other plants. Therefore, it is speculated that these novel miRNAs may be species-specific to wintersweet and are nonconserved miRNAs, whose expression requires certain induction conditions or have certain tissue specificity.

In the study, a comparative analysis of DEMs among IB570/FB.Nov, IB570/nF16, IB570/FB150, IB570/FB300, and IB570/FB450 was performed to identify miRNAs associated with the dormancy release of wintersweet floral buds, and 23 common DEMs were screened (Figure 3b). Among them, miR157-x, miR167-x, miR167-y, miR390-y, and miR395-y were highly conserved and downregulated after dormancy release. During the whole dormancy process, miR167-x showed high expression, whereas the expression of miR167-y was relatively low, and the other three miRNAs were also expressed at low levels. These miRNAs were also identified in other plant studies of floral bud dormancy, such as for pear and peach plants; however, their expression characteristics are not always consistent among plants. In white pear, two miR157 (Pyr-miR157a and miR157) and Pyr-miR390f were upregulated after floral bud dormancy release, whereas the expression trends of two miR167 (Pyr-miR167g and Pyr-miR167j) and Pyr-miR395i showed no significant difference throughout the dormancy process; moreover, during dormancy, these miRNAs, excluding Pyr-miR395i, showed high expression levels [21]. In the floral buds of the Japanese pear, miR395 was not identified, and the expression levels of miR157, miR167, and miR390 did not show significant differences between endodormancy and ecodormancy; only miR167 showed high expression in both stages [59]. In addition, miR167b and miR167c of the peach were highly expressed in dormant floral buds, whereas their expression was at a lower level after dormancy release [39]. These results suggest that miRNAs from the same family have different expression characteristics in different plants. Some scholars have proposed that different plants have different responses to the same cold stress [61]. Which indicates that the roles of miRNAs and the regulation mechanisms of floral bud dormancy differ among plants in their long-term evolution.

Identifying the target genes and their functions is an important way to analyze the biological functions and mechanisms of miRNAs. In plants, miRNAs mainly silence gene expression at the post-transcriptional level by targeting mRNAs for cleavage [43]. In this study, we identified 1707 target genes of 21 DEMs in wintersweet floral buds via degradome sequencing, which included 1940 miRNA–target pairs.

Of the 1707 target genes, a fraction were involved in flower development. And among these genes, a *FUL-like* homologous gene, *CpFUL-like*, was targeted by novel-m0128-3p (Appendix A). In our previous studies, *CpFUL-like* was regarded as a candidate gene for the regulation of wintersweet floral bud dormancy [41], and *35S::CpFUL-Like*/Col-0 transgenic Arabidopsis lines showed an early flowering phenotype compared with WT (unpublished data).

The *SVP/DAM* gene plays a central regulatory role in bud dormancy [21,22,23,24,25,26,27]. In white pear, miRNA6390 was found to directly target *PpDAM* and regulate the release of floral bud dormancy [21]. In our previous study, two *SVP* homologous genes (*CpSVP1* and *CpSVP2*) were identified from the transcriptome of wintersweet floral buds and considered as candidate genes for regulating floral bud dormancy [41]. However, miRNA6390 was not identified in all of the sRNA libraries in this study, and miRNAs targeting *CpSVP1* or *CpSVP2* were not found, which is similar to the results reported in Japanese pear, peach, and apple [37,39,59]. The reason for such differences may be different species, indicating that the regulatory roles of miRNAs in plant bud dormancy release may vary among species or even cultivars.

Plant hormones have been proved to be the most important internal factors regulating the dormancy cycle of deciduous trees [29]. Among them, the roles of ABA in dormancy regulation were most intensively studied, and numerous studies have shown that ABA plays a vital role in the establishment of bud dormancy [62,63,64,65,66]. In this study, of the 1707 target genes, some genes were involved in ABA biosynthesis and signal transduction, such as *zeaxanthin epoxidase* (*ZEP*), *protein phosphatase 2c* (*PP2C*), and *abscisic acid-insensitive 5* (*ABI5*) (Appendix A). ZEP is an important regulatory enzyme in the first step of ABA biosynthesis, and the content of endogenous ABA is closely related to bud dormancy process [29]. *PP2C*, considered to be the inhibitor, is a vital component of the ABA signal transduction pathway [29]. In pear, *PP2C* is expressed at a low level during bud dormancy and upregulated after the release of bud dormancy [65,67]. *ABI5* mediates ABA signal transduction. In our previous study, *ABI5* (Unigene0029724) was regarded as a candidate gene for the regulation of floral bud dormancy in wintersweet. Its expression was significantly downregulated with the accumulation of low temperatures [41], and it was targeted by novel-m0128-3p (Appendix A). Thus, it is speculated that miRNAs mediate ABA biosynthesis and signal transduction to regulate the dormancy release of floral buds in wintersweet.

Auxin, a major growth promoter, is involved in the regulation of dormancy release in many species. In tea plants, the content of endogenous IAA remains low during dormancy and increases continuously after dormancy release until spring germination [68]. In *Prunus mume*, the endogenous IAA level decreases as the floral buds enter dormancy and then increase continuously with the release of dormancy [69]. Furthermore, the transcriptome data of *Prunus mume*, Japanese apricot, and poplar show that the expression levels of auxin-associated genes are significantly different in buds between dormancy and dormancy release, and most of them are downregulated during dormancy [69,70,71]. However, for bud dormancy, the mechanism of auxin remains unclear. In this study, of the 1707 target genes, some genes involved in auxin signal transduction were found, including *ARF*, *IAA27*, *AUX22*, *SAUR36*, and *TIR1* (Appendix A). The auxin/indole-3-acetic acid (Aux/IAA) family, auxin response factor (ARF) family, and small auxin upregulated RNA (SAUR) are auxin response factors. Transport inhibitor response 1 (TIR1) is an auxin receptor and mediates transcriptional responses to auxin and the ubiquitination degradation of Aux/IAA [72,73,74]. The transcription factor ARF regulates the expression of auxin responsive genes and is regulated negatively by Aux/IAA [72,75]. These results indicate that miRNAs may regulate the dormancy release of floral buds by mediating auxin signal transduction in wintersweet.

Epigenetic regulation, such as DNA methylation and histone modification, is closely related to the induction, maintenance, and release of bud dormancy [18,30,31,32,33,34,35]. Here, the 1707 target genes contained multiple DNA methyltransferases and demethylases, and they also contained many enzymes involved in histone modifications such as histone methylation, acetylation, and ubiquitination (Appendix A), indicating that epigenetic modifications play important roles in the dormancy release of wintersweet floral buds. Among these genes, a histone acetyltransferase *HAT-B* and a histone deacetylase *HDA9* were predicted to be targeted by miR2092-x and novel-m0128-3p, respectively (Appendix A). Studies have shown that histone acetylation and deacetylation, which are catalyzed by histone acetyltransferases (HATs) and histone deacetylases (HDAs), respectively, play regulatory roles in cold responses in plants [76].

In addition, in this study, miR157-y and miR6293-x were found to target *RNA-dependent RNA polymerase 6* (*RDR6*) and *Dicer-like 4* (*DCL4*), respectively (Appendix A). *RDR6* and *DCL4* are mainly involved in the biogenesis of trans-acting siRNAs (ta-siRNAs) [77,78,79]. ta-siRNAs are a class of endogenous siRNAs with the length of 21 nt in plants, and they function similarly to miRNAs as post-transcriptional negative regulators [79,80]. It was reported that siRNAs could be involved in regulating the expression of MADS-box genes during bud dormancy transition in sweet cherry [35].

In this study, there were also a mass of target genes identified with unknown functions (Appendix A). Furthermore, degradome sequencing could not obtain the information that miRNAs regulate target genes by translational repression. Therefore, more studies will be needed to clarify the regulation of miRNAs in wintersweet floral bud dormancy.

## 4. Materials and Methods

### 4.1. Plant Materials and Total RNA Extraction

All plant materials used in this study were floral buds (FBs) of 18-year-old potted *C. praecox* ‘Concolor’ plants that were grown on the campus of Southwest University (106°43′ E, 29°83′ N, Beibei District, Chongqing City, China) with different chilling accumulation. Chill units (CU) were calculated according to the “UTAH Model” [81]. Plant treatment and sample collection were as described in our previous study (Materials and methods, pages 2–3) [41]. Samples included FB.Nov (FBs in November before treatment), nF16 (non-flowering FBs collected after treatment at 16 °C for −300 CU), FB150, FB300, FB450, and IB570 (FBs initiate blooming) (FBs collected after treatment at 12 °C for 150, 300, 450, and 570 CU, respectively). Total RNA was extracted from each sample using Trizol reagent (Invitrogen, Waltham, MA, USA) by following the manufacturer’s instructions.

### 4.2. sRNA Library Construction and Sequencing

After extraction of total RNA, RNA fragments of lengths 18–30 nt were enriched through polyacrylamide gel electrophoresis (PAGE) and were ligated with 3′ and 5′ adapters. Thereafter, reverse transcription and PCR amplification were performed, and 140–160 bp DNA fragments were enriched through agarose gel electrophoresis to generate the sRNA library [82]. The quality of the sRNA library was assessed using Agilent 2100 Bioanalyzer and qRT-PCR, and sequencing was performed using the Illumina HiSeq^TM^ 2500 platform by Gene Denovo Biotechnology Co. (Guangzhou, China).

### 4.3. Degradome Library Construction and Sequencing

RNA fragments with 3′ poly(A) were enriched from total RNA samples using magnetic beads with Oligo-d(T) and ligated with 5′ adapters. Then, the first strand of cDNA was reverse-transcribed using biotinylated random primers. Subsequently, PCR amplification was performed to generate the library [11,83,84,85]. Degradome sequencing was performed using the Illumina HiSeq 2000 platform by Gene Denovo Biotechnology Co. (Guangzhou, China) with the SE51 sequencing strategy to generate 50-nt reads.

### 4.4. Analysis of sRNA Sequencing Data

Raw reads from sequencing were filtered (removing contaminants, low-quality reads, and reads with length <18 nt reads) to obtain clean tags. All clean tags were mapped to the Genbank (Release 209.0, http://www.ncbi.nlm.nih.gov/genbank, accessed on 11 May 2019) [86], Rfam 11.0 (http://rfam.sanger.ac.uk/, accessed on 11 May 2019) [87], and RepeatMasker [88] databases to identify and remove non-coding structural RNAs (including rRNA, tRNA, snRNA, snoRNA, and scRNA) and possible repeat sequences. Subsequently, the clean tags were aligned with the *C. praecox* transcriptome database (NCBI accession number: PRJNA613935) established in our previous study [41] to remove the fragments from mRNA degradation. The remaining unannotated clean tags were used for miRNA identification.

The unannotated clean tags were aligned with miRBase 21.0 (http://www.mirbase.org, accessed on 11 May 2019) to identify known miRNAs in wintersweet samples with no more than two mismatches [89]. Thereafter, the remaining clean tags were mapped to the *C. praecox* transcriptome database to identify potentially novel miRNAs. Using default parameters, Mireap_v0.2 (http://sourceforge.net/projects/mireap/, accessed on 11 May 2019) was used to predict novel miRNAs and their hairpin structures [90,91]. The remaining tags were marked “unannotated”.

### 4.5. Analysis of Differentially Expressed miRNAs

Based on the abundance of miRNAs in each sample, the miRNA expression level was calculated and normalized to transcripts per million (TPM) with the following formula: TPM = actual count of miRNA tags/total count of clean tags × 10^6^ [11]. Following normalization, if the expression level of a given miRNA was zero, then the expression value was adjusted to 0.01 for further calculation [44,90]. Differentially expressed miRNAs (DEMs) were identified with edgeR (using default parameters) [92] with the criteria of $|\log_{2}\left(\mathrm{FC}\right)|\;\geq \;1$ and *p*-value < 0.05. Fold change (FC) represents the ratio of the expression level between two samples. FC ≥ 2 and FC ≤ 0.5 represent upregulated and downregulated expression, respectively. 

### 4.6. Analysis of Degradome Sequencing Data and Identification of miRNA Targets

After degradome sequencing was complete, adapters, contaminants, and low-quality reads were removed from raw reads to obtain clean tags. Then, non-coding structural RNAs were removed from clean tags as done for the sRNA-seq data. Degraded fragments with a single base ratio > 0.7 were defined as polyN, which were also removed from clean tags. The remaining clean tags were aligned to the *C. praecox* transcriptome database using the SOAP 2.20 (http://soap.genomics.org.cn/, accessed on 28 May 2019) [93] software to detect potentially degraded target genes. The tags mapped to the transcriptome were annotated as cDNA (cDNA_sense and cDNA_antisense) and used for the prediction of miRNA cleavage sites. CleaveLand 3.0 (http://axtell-lab-psu.weebly.com/cleaveland.html, accessed on 28 May 2019) [94] was used to count cleavage sites and build degradome density files. PAREsnip (http://srna-workbench.cmp.uea.ac.uk/tools/paresnip/, accessed on 28 May 2019) [95] was used to detect miRNA–target pairs with the criterion of *p*-value < 0.05. Finally, the target plot (T-plot) of miRNA–target pairs was constructed based on the abundance and position of degradome density files with CleaveLand 3.0. The process of degradome sequencing is shown in Appendix A. In addition, Gene Ontology (GO) enrichment analysis was performed for the identified target genes of miRNAs. The target genes were mapped to the GO database (http://www.geneontology.org/, accessed on 28 May 2019) using the GOseq R package, and the number of genes for each term was calculated. Then, the obtained statistical results were subjected to hypergeometric testing, and GO terms with *p*-value ≤ 0.05 were considered significantly enriched [96].

miRNA–target pairs were grouped into five categories according to the abundance of tags at the cleavage site with decreasing reliability [97]: category 0, the abundance of tags at this cleavage site is only the maximum on the transcript; category 1, the abundance of tags at this cleavage site and others is tied for the maximum on the transcript; category 2, the abundance of tags at this cleavage site is higher than the median but not the maximum on the transcript; category 3, the abundance of tags at this cleavage site is equal to or less than the median on the transcript; category 4, only one tag is found at this cleavage site.

## 5. Conclusions

This is the first study to analyze and identify the miRNAs and their target genes that regulate the dormancy release of floral buds in wintersweet using high-throughput sequencing technology. As a result, a total of 862 known and 402 novel miRNAs were identified with sRNA sequencing. The length of their sequences ranged from 18 nt to 26 nt, with 21-nt miRNAs being the most abundant (85, 50%), and the nucleotide at the 5′ terminal of these miRNA was mainly biased toward U or A. Moreover, 23 DEMs (10 known and 13 novel) were screened by comparing the breaking and other dormancy samples, which indicated that miRNAs are involved in regulating the dormancy release of wintersweet floral buds. In addition, 1707 target genes of 21 DEMs were identified with degradome sequencing, and the potential functions of these targets were also annotated and analyzed preliminarily. The results provide valuable reference material for further elucidating the regulation of miRNAs in the dormancy release of wintersweet floral buds.

## Figures and Tables

**Figure 1 ijms-24-04210-f001:**
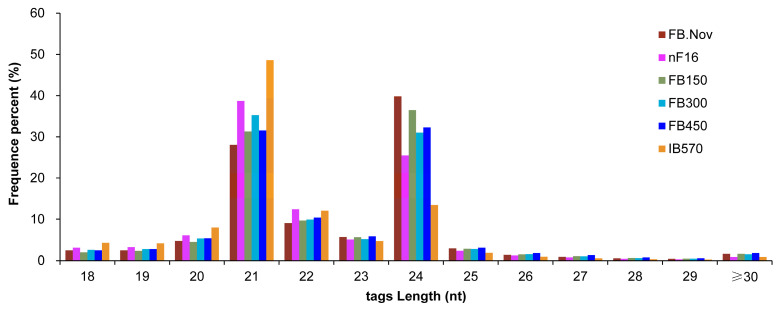
Sequence length distribution of sRNAs in the floral buds of wintersweet. FB.Nov: FBs in November before treatment; nF16: FBs collected after treatment for −300 CU chilling accumulation at 16 °C; FB150, FB300, FB450, and IB570: FBs collected after treatment for 150, 300, 450, and 570 CU chilling accumulations at 12 °C, respectively.

**Figure 2 ijms-24-04210-f002:**
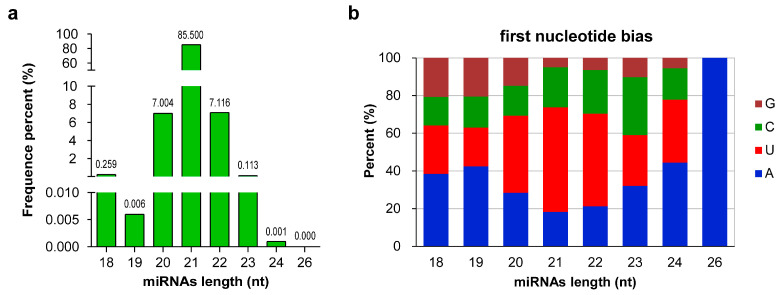
Sequence information of all microRNAs (miRNAs) in wintersweet floral buds. (**a**) Sequence length distribution of miRNAs. The numbers in the figure represent the percentage of miRNAs sequences with different lengths in the total miRNA sequences. (**b**) First base distribution of miRNAs of different lengths. A, U, C, and G represent adenine, uracil, cytosine, and guanine, respectively. (**a**,**b**) were generated with GraphPad Prism 9 software [48].

**Figure 4 ijms-24-04210-f004:**
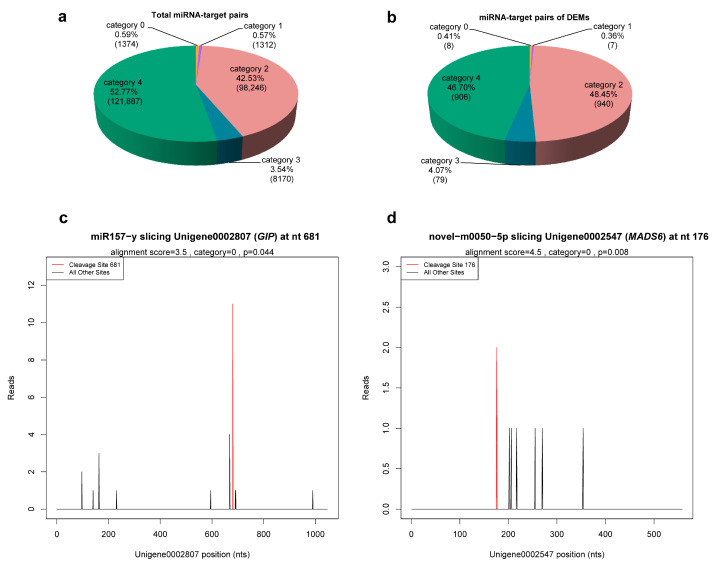
miRNA–target pairs. (**a**,**b**) Classification statistics of miRNA–target pairs. Numbers in parentheses indicate number of miRNA–target pairs. (**c**,**d**) The T-plots of two target genes confirmed by degradome sequencing. The T-plots were completed with CleaveLand 3.0.

**Figure 5 ijms-24-04210-f005:**
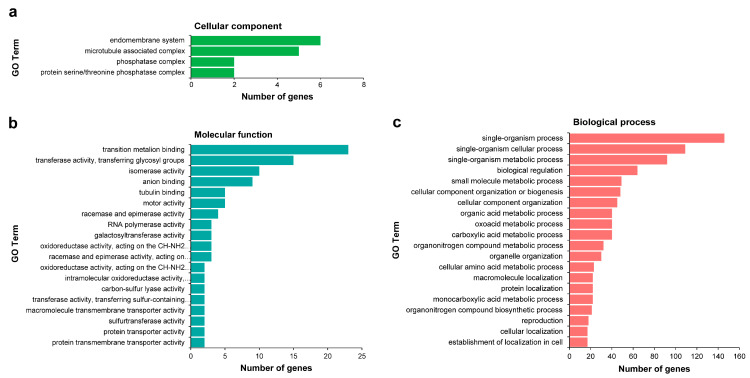
Gene Ontology (GO) enrichment analysis of differentially expressed miRNAs (DEMs) target genes. The figure shows significantly enriched GO terms (*p*-value ≤ 0.05) in three categories: Cellular component (**a**), Molecular function (**b**), and Biological process (**c**). All of the GO terms are contained in (**a**,**b**); the twenty most dominant GO terms are contained in (**c**). GO analysis was performed with the GOseq R package. The results were visualized on level 3.

**Table 1 ijms-24-04210-t001:** Annotation and classification of small RNAs.

Category	Count of Each Sample						Total	
	FB.Nov	nF16	FB150	FB300	FB450	IB570	Count	Percent (%)
Clean tags	59,744,323	48,982,971	60,793,889	54,640,035	54,201,813	46,146,283	324,509,314	100.00
rRNA	5,502,518	7,399,048	5,744,220	5,189,114	5,899,625	5,018,955	34,753,480	10.71
snRNA	100,945	91,590	65,898	58,395	66,318	108,220	491,366	0.15
snoRNA	39,384	58,075	51,357	58,198	60,944	50,743	318,701	0.10
tRNA	219,683	183,278	307,234	312,200	285,118	276,611	1,584,124	0.49
scRNA	0	0	0	0	0	0	0	0.00
Repeat	0	0	0	0	0	0	0	0.00
Transcriptome	13,462,284	11,230,416	13,953,551	12,197,013	13,505,105	9,878,030	74,226,399	22.87
Known miRNAs	10,771,326	9,898,793	12,640,126	13,364,950	11,784,001	15,704,868	74,164,064	22.85
Novel miRNAs	167,975	59,549	226,927	155,615	244,083	160,774	1,014,923	0.31
Unannotated	29,480,208	20,062,222	27,804,576	23,304,550	22,356,619	14,948,082	137,956,257	42.51

The count in the table were statistics for redundant tags. FB.Nov: FBs in November before treatment; nF16: FBs collected after treatment for −300 CU chilling accumulation at 16 °C; FB150, FB300, FB450, and IB570: FBs collected after treatment for 150, 300, 450, and 570 CU chilling accumulations at 12 °C, respectively. Transcriptome: degraded fragments.

**Table 2 ijms-24-04210-t002:** Annotation and classification of degradome data.

Category	Total Tags		Unique Tags	
	Count	Percent (%)	Count	Percent (%)
Clean tags	37,913,833	100.00	12,290,544	100.00
rRNA	542,685	1.43	21,435	0.17
tRNA	488	0.00	140	0.00
snRNA	762	0.00	320	0.00
snoRNA	26,782	0.07	371	0.00
polyN	65,227	0.17	32,771	0.27
cDNA_sense	14,139,730	37.29	3,850,879	31.33
cDNA_antisense	12,076,616	31.85	3,581,528	29.14
Unannotated	11,061,543	29.18	4,803,100	39.08

## Data Availability

Not applicable.

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
