# Peer review of "Small RNA and Degradome Sequencing in Floral Bud Reveal Roles of miRNAs in Dormancy Release of Chimonanthus praecox"

_ijms, 2023, doi:10.3390/ijms24044210_

Round 1

Reviewer 1 Report

The research paper titled, “Small RNA and degradome sequencing in floral bud reveal roles of miRNAs in dormancy release of Chimonanthus praecox” reports important data about the miRNAs that have the roles of floral bud dormancy release in Chimonanthus praecox.

However, the paper needs some of review and answer following comments.

Line 74. The full form of the abbreviation “H3K4me3, H3K27me3, and H3ac” should be stated.

Line 96. “sometimes” not true for academic language.

Line100-101. However, the roles of miRNAs involved in chilling-mediated dormancy control of wintersweet floral buds remain unclear.” connect with the previous sentence

Line101-102. Please cite the reference stated regarding “High-throughput sequencing technology and bioiformatics methods are effective methods to identify miRNAs.”

Line 106. The sentence starting with “The study provides” is the same with the sentence in the summary and conclusion parts. Please make them unique.

Results

An explanation should be added for all of the figures and it should be stated in which program they were created according to which criteria.

The sample groups taken for the sequencing given in the method (line 455) are not given clearly and the sequencing comparisons between the sample groups taken are not specified separately in the results. For example, it is not clear which sample group the Table 1 contains. The table should be rearranged according to a sample groups. The results need to be reviewed accordingly.

Line 116. It's better to say “358 million clean reads” instead of “358,543,196 clean reads”. Others should be corrected accordingly.

Line 119. Spelling error should be corrected.

Line 121-122. Line errors should be corrected.

Line 129. In Figure 1, the full names of the abbreviations corresponding to the colorings should be given.

 Line 137. The numbers and percentages in Table 1 are not specified to which application group they belong. The table is not clear.

Line 174.  Figure 2 can be given as supplementary data

Line 225. Figure writings are not clear. All manuscripts should be checked

Line 266. Please change the color of a and b of figure  5.

Line 290. It should be specified in which program and under which conditions the figures were created. All manuscripts should be checked

The summary of the research results of this article could be better. The author is recommended to summarize the main results of this article with (1), (2), (3).

Line 406. Style used in naming genes should be consistent all over the manuscript, use the Italic form for all genes.

Materyal & Metod

Please create a figure to summarize the process of the degradome sequencing Chimonanthus praecox from collecting samples to analyzing sequencing data.

Line 453. It should be stated where the plant material was obtained from. The place provided in the previous work cited is not specified.

Line 477, 484, 488. Authors should cite literature of all software and webtools used in the material method section in the manuscript (Genbank, Rfam, and RepeatMasker, miRBase , Mireap_v0.2, etc.).

Line 490. Please correct the spelling error “unann”.

Line 498. |log2(FC)|≥1 threshold is important but the threshold can be selected higher in other studies

Line 513-514. It should be stated with which software or web tool the analysis of Gene Ontology (GO) enrichment was performed.

Reviewer 2 Report

Liu et al., studied the roles of microRNAs in floral bud dormancy using sRNA-seq and degradome sequencing in Chimonanthus praecox. The authors identified over 1,000 known and novel miRNAs and 23 DE miRNAs during bud dormancy release. The target genes of these DE microRNAs suggested the biological processes that may participate in the bud dormancy of wintersweet. The manuscript is well organized and written, but there are some major issues I wanted to point out.

Major concerns:

1. Line 154, the sentence is not true about lacking chimonanthus praecox genome data. Two genomes were published in 2020 (Shang, J., Tian, J., Cheng, H. et al. The chromosome-level wintersweet (Chimonanthus praecox) genome provides insights into floral scent biosynthesis and flowering in winter. Genome Biol 21, 200 (2020). https://doi.org/10.1186/s13059-020-02088-y) and in 2021 (The red flower wintersweet genome provides insights into the evolution of magnoliids and the molecular mechanism for tepal color development).  The authors should consider using the genome for microRNA annotation instead of a transcriptome. 

2. The authors should apply some filtering for the degradome-supported gene targets. In Figure 5A, half of the cleavage sites were in the Cleaveland Category 4, which means they have only one sequence evidence. For stringent target identification, targets in Category 3 and 4 are not reliable and should be filtered out. I would recommend the author filter the miRNA targets and re-do the GO enrichment analysis.

Minor concerns:

1. The authors described the experiment well in the methods section but it is at the end. To make the manuscript read smoothly, it would be great to briefly describe the experimental design at the beginning of the result including: What wintersweet cultivar is used? What is its chilling requirement? Briefly explain the six time points and the reason for choosing these six time points.

2. Chimonanthus as a model plant in the magnoliids clade, which is phylogenetic distant to the Rosaceae family, how similar of the wintersweet bud dormancy process compared to those fruit species? I am asking this because the authors compared candidate genes and pathways that were extensively studied in Eudicots. What if the wintersweet is totally different from them? What the authors found might be specific in the magnoliids, which provides insights into bud dormancy mechanisms in a diverse way.  

Round 2

Reviewer 1 Report

Thanks for your revision. 

Author Response

We gratefully thank you for the constructive remarks and useful suggestions, which has significantly raised the quality of our manuscript. At the same time, your comments made us gain a lot and will have important guiding significance for our future research and article writing. Thank you again sincerely.

Reviewer 2 Report

Thank you for the responses, the authors addressed most of my comments but there are some new issues when I went through the manuscript again.

1. The authors explained to me very well why transcriptome is used for miRNA annotation, but did not talk about it at all in the manuscript. Please explain it in the manuscript why the transcriptome database is used instead of the two genomes.

2. The statistical analysis for GO enrichment is not described. The authors performed a GO enrichment on the miRNA target genes, the main figure should present only the statistical significant GO terms instead of all of them. And please provide detail about your GO enrichment analysis in method: what statatistical test did you use?  What threshold/cutoff do you use to determine significance? 

And the level 2 GO term is really vague and general, it would be interesting to identified more specific biological processes or pathways. Did the author identified significant GO terms in higher levels?

3. Figure 2 is confusing. 2a showed 0% of 24 and 26-nt miRNAs, but 2b had their nucleotide bias. Where did those numbers come from? There is no method describing how figure 2 is generated. 

I am also confused about that the authors showed more than 30% of sRNA tags in 24nt in Figure 1, and none of them were annotated as miRNA. Please explain a bit what those 24nt small RNAs are? 

4. Line 219, the authors specifically pointed out that two miRNS novel-m0186-3p and novel-m0077-5p may be responsive to chilling and involved in dormancy regulation, but further discussion is missing for them. Would it be interesting to see what genes/pathways they are targeted? 

5. Conclusions section is the same as abstract, which makes it feel like redundant, and weaken the manuscript. I would suggest the authors rewrite the entire conclusion, to include more detailed findings from the study, and your perspectives of the miRNA regulation in wintersweet dormancy.

6. Typo: line 172, this sentence has two verbs, 'identified revealed'.

Round 3

Reviewer 2 Report

I would like to thank the authors for addressing my comments. The manuscript is improved and I do not have further questions.